# Artificial Intelligence-Driven Penetration-Aspiration Detection in Dysphagia Patients Using Fluoroscopic Videos

Sanjeevi G*, Uma Gopalakrishnan*, Rahul Krishnan Pathinarupothi*,
Arya CJ†, Kaustuv M Talukdar‡, and Subramania Iyer K†
*Center for Wireless Networks & Applications (WNA), Amrita Vishwa Vidyapeetham, Amritapuri, India
†Department of Head and Neck Surgery, Amrita Institute of Medical Sciences and Research Center, Kochi, India
‡Department of Head and Neck Surgery and Oncology, Amrita Institute of Medical Sciences and Research Centre, Faridabad

*Abstract*—The gold standard for diagnosing dysphagia is the Videofluoroscopic Swallowing Study (VFSS). In patients with dysphagia, the invasion of food material into the airway is known as penetration-aspiration. Assessing this risk using VFSS is inherently subjective, with significant inter-patient and inter-rater variability. This article proposes an AI pipeline that introduces a novel approach in which bolus segmentation and airway detection are combinationally assessed to interpret frame-wise penetration-aspiration risk. The existing AI approaches rely on manual frame selection and overlook the clinical significance of bolus and airway. Additionally, addressing challenges posed by varying airway orientations, we develop an automated AI pipeline that tracks bolus and airway throughout VFSS videos. We curated a VFSS dataset and annotated one-third of the frames from 82 VFSS clips obtained from 40 patients due to a lack of benchmarks. Our approach involved comparing various segmentation models for bolus segmentation and fine-tuning object detection model for airway detection. The segmented bolus area and airway information are then processed to identify penetration-aspiration events. Our pipeline achieved a dice score of 0.80, a mean average precision of 0.93, and an accuracy of 89% in bolus segmentation, airway detection, and penetration-aspiration detection. Our pipeline could be effectively trained even with limited annotated frames. This saved clinicians time and also reduced the burden of manual annotation. These promising results have significant potential for assisting clinicians in assessing penetration-aspiration risk.

*Index Terms*—Dysphagia, Airway detection, Bolus segmentation, Deep learning

## I. INTRODUCTION

Dysphagia, a disorder characterized by difficulty swallowing, is often associated with physiological impairments that manifest during eating and drinking and is a common condition affecting approximately 43% of the population [2], [3]. The significant medical concern of dysphasia is airway invasion, a pathological process that can lead to pneumonia or asphyxia. Due to this invasion, food may aspirate (enter the airway below the vocal fold) or penetrate (touch the vocal fold). Aspiration pneumonia is a common cause of illness and death among the elderly. Both healthy people and patients with an inadequate cough reflex have small-volume aspiration of oropharyngeal secretions while they sleep. On the other hand, people who have dysphagia as a result of diseases such as stroke, dementia, head and neck malignancies, and esophageal motility problems experience large-volume aspiration [4], [5].

The videofluoroscopic swallowing study (VFSS) is the gold standard for dysphagia assessment. It is an x-ray modality that records the swallowing process of patients with different food consistencies. The VFSS videos primarily encompass a sagittal view of the oropharynx, pharynx, and upper esophagus, capturing all pertinent swallowing structures [1]. This enables clinicians to assess the physiological function of the swallowing process by visualizing bolus flow and structural movement throughout the upper aerodigestive tract, aiding in determining the severity of the condition. The VFSS images are complex and require careful frame-by-frame analysis, which can be time-consuming and subjective. The airway invasion assessment using VFSS involves subjectivity and is time-consuming, and there are reported cases of inter- and intra-rater variability [5].

Artificial intelligence (AI) has been utilized widely in medical image analysis for computer-aided diagnosis. Notably deep learning (DL) algorithms are predominantly used for classifying abnormality, segmentation of structure, and tracking objects in medical images. Researchers also tried the application of DL in VFSS analysis. This includes pharyngeal phase detection, pharyngeal swallow reflex detection, bolus segmentation, hyoid bone detection, and temporal parameter analysis [5], [8], [9]. There is a recent interest in the automatic assessment of airway invasion from VFSS. The studies have tried AI, specifically DL algorithms to detect the penetration-aspiration condition [16]–[18]. These studies are limited to manual frame selection, not considering the significance of bolus flow and airway in the VFSS images. These studies identify the risk of penetration-aspiration directly from VFSS images. However, in this scenario, there is no guarantee that the model predicts the penetration-aspiration based on the features of the bolus and airway. Therefore, we are motivated to create an AI pipeline that includes bolus segmentation and airway detection for airway invasion diagnosis.

Bolus segmentation helps to accurately track the movement

of the bolus during swallowing, which is crucial for detecting penetration and aspiration. Airway detection, particularly identifying anatomical structures like the vocal cords and glottis, provides interpretable insights into the AI's decision-making process. In combination, bolus segmentation and airway detection provide the AI model with a more complete understanding of the swallowing process [26], [27]. This can help the model to more accurately detect penetration and aspiration events, and expedite the rating of VFSS recordings. However, achieving this goal has the following research challenges.

The irregular shape of the bolus in VFSS poses challenges for segmentation annotation. Additionally, the orientation of the airway varies among VFSS images due to patient-specific differences. To accurately identify the airway, including the vocal fold and larynx, we annotate it as an oriented bounding box with an angle. However, fine-tuning an object detection model to predict the oriented bounding box is challenging because existing models typically predict only regular rectangular bounding boxes. Penetration-aspiration events are then determined using information from both the segmented bolus and airway areas. Specifically, we calculate the overlap between the segmented bolus mask and the airway box. In VFSS clips, temporal alignment between bolus segmentation and airway detection is crucial for frame-wise risk assessment. Setting a threshold for overlap and aggregating the frame-wise risk remains challenging when categorizing normal, penetration, or aspiration cases in swallowing clips. In this article, we develop an automated AI pipeline that includes the bolus and airway tracking throughout the VFSS video and detects airway invasion. We have examined the different state-of-the-art DL algorithms in bolus segmentation and airway detection. The combination of these modules improves the airway invasion diagnosis.

The following are the main contributions of the article:

- To the best of our knowledge, for the first time, we employ a deep-learning model to detect airway in VFSS for automatic airway invasion detection.
- We introduce an AI pipeline specifically designed for airway invasion detection. The pipeline comprises three key components:
  - Bolus Segmentation Model: Identifies and segments the bolus (swallowed food material) in VFSS videos.
  - Airway Detection Model: Detects the airway region within the VFSS frames.
  - Decision-Making Module: Integrates the bolus and airway information to make informed decisions. Experimental results demonstrate that our pipeline achieves precise tracking of both bolus and airway, yielding comparable outcomes.
- To facilitate research in this domain,
  - We curate the dataset for airway detection, bolus segmentation, and penetration-aspiration detection. Our dataset comprises 82 VFSS videos from 40 subjects.
  - We annotate ground truth labels for bolus and airway

boxes in every one-third of frames within the VFSS clips. We observed this limited number of frames is sufficient to train the DL models for bolus segmentation and airway detection. Our results align with state-of-the-art methods for bolus segmentation. Importantly, this annotation process significantly reduces the time burden on swallowing pathologists.

This article is structured as follows: we address related works in Section II, which includes DL-based VFSS analysis. The VFSS dataset and our methodology are described in Section III. The implementation details, experiments, and results are presented in Section IV. We analyzed the results and compared them with previous studies in Section V. Section VI concludes the article.

## II. Related Works

### A. Bolus segmentation

During VFSS, the direction of flow and placement of the bolus are critical factors in diagnosing dysphagia. Clinicians can observe the dynamic bolus flow in real time using VFSS. The constantly shifting perceptual features of the bolus during transit make frame-by-frame quantitative analysis of bolus flow subjective, labor-intensive, and time-consuming, even if VFSS permits monitoring of the bolus's velocity, and trajectory during swallowing. Addressing the need for automated bolus detection, Caliskan et al. [10] employed a DL framework, specifically a Mask-RCNN, to automate the identification of bolus in videofluoroscopic image frames. Building upon this, Ariji et al. [11] and Shaheen et al. [12] developed a DL model for automated bolus segmentation on VFSS images using a U-Net neural network. In parallel, Bandini et al. [6] implement weakly supervised learning for bolus localization in VFSS by using a DL framework. The approach leverages convolutional neural network (CNN) and class activation maps to identify the pharyngeal phase and localize the bolus without manual annotation of bolus location.

Additionally, Li et al. [13] evaluated various DL models for VFSS bolus segmentation. The InceptionResNetV2 encoder in the UNet++ architecture performed best among other models. Zeng et al. [14] introduced Video-TransUNet, a deep architecture for segmenting the bolus and pharynx. The method combines a Vision Transformer for non-local attention, a ResNet CNN backbone for strong frame representation, a UNet-based convolutional-deconvolutional architecture with multiple heads for reconstructive capabilities for multiple targets, and multi-frame feature blending for multi-targets. Zeng et al. [15] proposed Video-SwinUnet for segmentation in VFSS in a separate study. The methodology collects features from nearby frames throughout the temporal dimension and merges them using a temporal feature blender. The high-level spatiotemporal feature is subsequently tokenized, and the final segmentation results are generated using an encoder-decoder architecture similar to UNet.

## B. Penetration-Aspiration Risk Assessment

Airway invasion, a pathological process that may result in pneumonia or asphyxia, poses a significant clinical concern. The consequences of airway invasion include airway obstruction, pneumonia, and asphyxia. Therefore, identifying patients at risk of penetration-aspiration is crucial. However, assessing airway invasion using VFSS involves subjectivity and is time-consuming, with reported cases of variability among clinicians [5]. To address this, several researchers have proposed DL frameworks. For instance, Lee et al. [16] developed a deep-learning pipeline to detect airway invasion from videofluoroscopic videos. The extended study measured inter and intra-rater variability for the DL model and humans in penetration-aspiration detection. The DL model from the study [16] and three clinicians evaluated VFSS videos to determine whether or not penetration-aspiration was present. The outcomes show that the DL algorithm is as accurate as human examiners.

Building on this, Reddy et al. [18] sought to identify the most effective DL architecture for detecting penetration-aspiration from VFSS. They compared 2D-CNN, 3D-CNN models, and CNN-LSTM, demonstrating the superiority of 3D-CNN in VFSS classification and the efficiency of multi-label classification for 2D-CNN models. On top of this, Sanjeevi et al. [7] introduced SPAD, a clinical tool that classifies different phases of swallowing and identifies penetration-aspiration by assessing bolus residue. DL methodologies have shown promising results in bolus segmentation and penetration-aspiration detection. However, these methodologies face limitations for the following reasons: The clinical significance of bolus segmentation and tracking remains to be explored. Existing methods for penetration-aspiration involve manual frame selection and ROI determination, which limits automated airway invasion detection in real-time practice.

## III. METHODOLOGY

This article presents an AI pipeline designed for detecting penetration-aspiration in VFSS clips. Our pipeline consists of two key modules: the bolus segmentation module and the airway detection module as illustrated in Figure 1. To create a robust dataset, we collected VFSS swallowing clips and annotated them for bolus segmentation, airway detection, and penetration-aspiration detection. We then evaluated various state-of-the-art segmentation models to assess their performance in bolus segmentation. We proposed the YOLO-V8 model for airway detection, which has demonstrated exceptional accuracy. Our pipeline works as follows: i). Data Extraction: We extracted VFSS image frames from the swallowing clips. ii). Preprocessing: We applied contrast-limited adaptive histogram equalization (CLAHE) to enhance the image quality of all extracted frames. iii). Bolus segmentation and airway detection: The preprocessed images serve as input for our segmentation models. These models perform bolus segmentation, while YOLO-V8 focuses on detecting the airway in VFSS images. The predicted bounding box coordinates from YOLO-V8 are used to create rectangular mask images for the airway. iv). Decision-Making Module: We process both the

TABLE I: Patient Demographic details

| Characteristics | Value |
|---|---|
| Age, Year | 60.69 ± 15.63 |
| Sex, Male: Female | 25:15 |
| Class, Normal: Penetration-Aspiration | 40:42 |
| Stroke | 6 |
| Brain Tumor | 4 |
| Cancer | 6 |
| Neurological disorder | 10 |
| Parkinson disease | 3 |
| Heart disease | 4 |
| Neck Surgery | 7 |

bolus image and the airway rectangle images in our decision-making module for airway invasion detection.

## A. VFSS Dataset

This study analyzed VFSS video recordings from 40 subjects experiencing swallowing difficulties. The VFSS swallow data was collected from patients admitted between 2021 and 2023 at Amrita Institute of Medical Sciences and Research Center, Kochi, with ethical approval from the Institutional Review Board. Informed consent was secured from all study participants. During VFSS procedures, subjects were positioned upright before a video fluoroscopic device, capturing a lateral view of the neck and head. Diluted barium was mixed with two different substances - 3 mL and 5 mL of thick and thin liquid, respectively. Some subjects struggled to fully swallow the substances due to significant aspiration or delayed reflexes. 82 video clips (recorded at 8 frames per second) were collected, each featuring one swallow instance during the VFSS. Video clip durations ranged from 10 to 15 seconds. Population class distribution and demographic details are summarized in Table I.

*1) Ground Truth Labeling:* The ground truth labels for penetration-aspiration risk, bolus segmentation, and airway detection were established by two deglutologists and reviewed by an expert physician. For airway invasion detection, annotations were applied to 82 VFSS swallowing clips, categorizing them as either 'normal' or 'penetration-aspiration.' The airway invasion labeling utilized the penetration-aspiration scale, where scores of one indicate normal swallowing, while scores of two to eight indicate airway invasion [18]. Bolus segmentation and airway detection annotations were performed at the frame level. A total of 7,068 frames were extracted from the 82 swallowing clips. In VFSS videos, adjacent frames exhibit high similarity. However, significant changes occur in VFSS components in approximately every one-third of the frames. To reduce annotation workload, clinicians annotated bolus masks and airway bounding boxes in every one-third of frames within VFSS video clips. The airway region in VFSS images was annotated using oriented rectangle bounding boxes. The CVAT annotation tool [28] was used for annotating the VFSS frames. Overall, 2,356 frames received bolus mask and airway bounding box annotations.

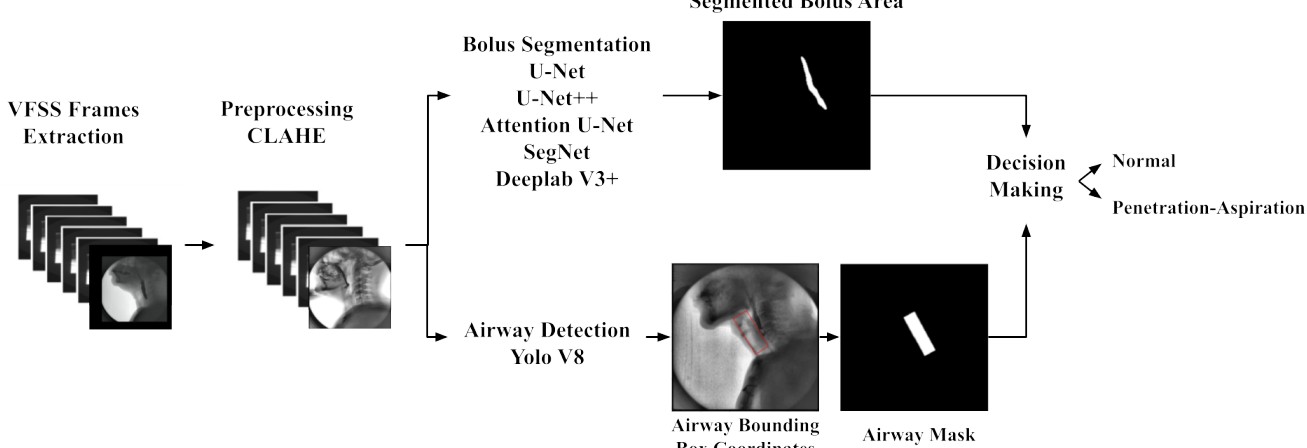

Fig. 1: Illustration of methodology: VFSS image frames are extracted and enhanced using CLAHE. Segmented bolus and the airway, aiding airway invasion detection.

## B. Bolus Segmentation

The bolus travels from the oral cavity to the stomach and is crucial in airway invasion detection. We enhance bolus tracking and facilitate informed decision-making by accurately segmenting the bolus in VFSS images. CNN-based models have emerged as the predominant choice for medical image segmentation. Our study explored various CNN segmentation algorithms, such as U-Net, U-Net++, Attention U-Net, DeepLab V3+, and SegNet. These models were rigorously evaluated across diverse medical image modalities and consistently demonstrated state-of-the-art performance in medical image segmentation. We identified the optimal model to enhance penetration-aspiration prediction by comparing their results.

*1) Segmentation Models:* The U-Net architecture consists of an expanding path (decoder) and a contracting path (encoder). The encoder uses 3 × 3 convolutions with ReLU activation and 2 × 2 max pooling for downsampling, while the decoder uses 2 × 2 up-convolutions. Skip connections link the encoder and decoder, allowing fine-grained features to influence image generation. The final layer converts multi-channel features into a single-channel mask with 1 × 1 convolutions and sigmoid activation [21]. UNet++ enhances U-Net by integrating convolution blocks into skip connections, eliminating the need to trim and copy features. Dense convolution blocks and deep supervision during training further improve learning [22].

Attention gates (AGs) in U-Net reduce computing overhead and enhance prediction accuracy by focusing on relevant features. Skip connections combine coarse- and fine-level predictions and multi-scale feature extraction captures contextual information [23]. DeepLabv3+ uses atrous convolutions in the encoder to capture multi-scale context and atrous Spatial Pyramid Pooling to enhance feature capture. The decoder blends low-level and coarse semantic features, using atrous separable convolutions to preserve spatial information [24].

SegNet's encoder extracts features using convolutional layers and max-pooling blocks, while the decoder uses max-pooling indices for non-linear up-sampling, recovering spatial information. The final layer is a sigmoid classifier for pixel-by-pixel classification [25].

## C. Airway Detection

The airway plays a critical role in assessing the risk of penetration and aspiration. Analyzing the airway component within VFSS images is essential for evaluating this risk. Additionally, tracking the airway in VFSS images contributes to accurate penetration-aspiration risk assessment. In the field of computer vision, YOLO (You Only Look Once) family networks are commonly used for object detection [32]. In VFSS images, the airway appears as an oriented object. Leveraging the state-of-the-art YOLO-V8 model, which supports oriented bounding box predictions [33], we fine-tuned it specifically for airway detection in VFSS images.

*1) Yolo-V8:* YOLO-V8, an algorithmic model, was created by Ultralytics [29]. The head, neck, and backbone are the three primary parts of its network architecture. Updated CSPDarknet53 serves as the backbone network for YOLO-V8. Five separate scale features are obtained by downsampling input features five times. With the C2f module, the original Cross Stage Partial (CSP) module in the backbone is replaced. By using gradient shunt connections, the C2f module improves information flow without sacrificing design. Convolution is applied to the input data by the CSP module, which then produces the output by batch normalization and SiLU activation. To adaptively pool input feature maps to a fixed-size output, the backbone network integrates the spatial pyramid pooling fast (SPPF) module [19], [20].

Neck: YOLO-V8 has a PAN-FPN (Pyramid Attention Network-Feature Pyramid Network) structure in the neck, which was inspired by PANet (Path Aggregation network). Through the combination of top-down and bottom-up networks, PAN-FPN improves feature completeness and variety

by combining deep semantic information with shallow location information. Head: A decoupled head structure is used by YOLO-V8's detection component. Separate branches for predicted bounding box regression and classification of objects are included in this framework. Both of these tasks are applied with different loss functions. Positive and negative samples are properly defined by the anchor-free detection model. To fine-tune the head portion, we froze the YOLO-V8 backbone and neck weights in this study. To create the mask on the airway of VFSS frames, our objective was to determine the airway and forecast the orientated bounding box coordinates [19], [20].

### D. Penetration-Aspiration Detection

The frames were extracted from VFSS video clips. After preprocessing, these frames serve as input to the segmentation and airway detection model. The bolus information and the airway bounding box coordinates in VFSS images are extracted from the segmentation model and YOLO-V8, respectively. Subsequently, a binary mask for the bounding box is generated based on predicted coordinates. Next, we applied a logical AND operation between the predicted bolus masks and the airway bounding boxes. If overlapping pixels occur, the frame is classified as a penetration-aspiration frame; otherwise, it is considered normal. This frame-level classification applies to each frame in the VFSS clip. Additionally, we leveraged the efficient Boyer-Moore algorithm [30] which is an efficient algorithm for finding the majority element in a list. Specifically, the Boyer–Moore majority vote algorithm was employed to aggregate the VFSS swallowing clip into either normal or penetration-aspiration categories.

## IV. EXPERIMENTS & RESULTS

### A. Implementation

The experiments were conducted in computational environments hosted on a cloud server equipped with an Intel® Xeon® processor and an Nvidia V100 GPU. All experiments were performed using Python 3.10.12 and the PyTorch 2.3.0 framework.

### B. Training

To reduce computational complexity, all images were resized to a resolution of 224x224x3. The segmentation models were initialized with ImageNet pre-trained weights in the encoder part, using ResNet50 as the backbone. We employed the AdamW optimizer and a loss function combining binary cross-entropy and dice loss. For all models, training lasted 100 epochs, with early stopping and lower learning rate adjustments. Additionally, we fine-tuned the YOLO-V8 model pre-trained on the DOTA v1 dataset [31] for 50 epochs, using a batch size of 8 for airway-oriented bounding box prediction on our dataset. The loss function combined distribution focal loss and complete intersection over union (CIoU), and we used the AdamW optimizer. To avoid overfitting the data augmentation on images such as random rotation and flipping were carried out.

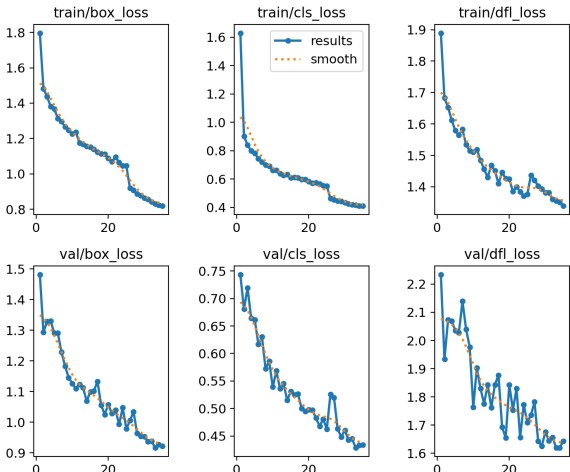

Fig. 2: Loss convergence of Yolo-V8 model in airway detection.

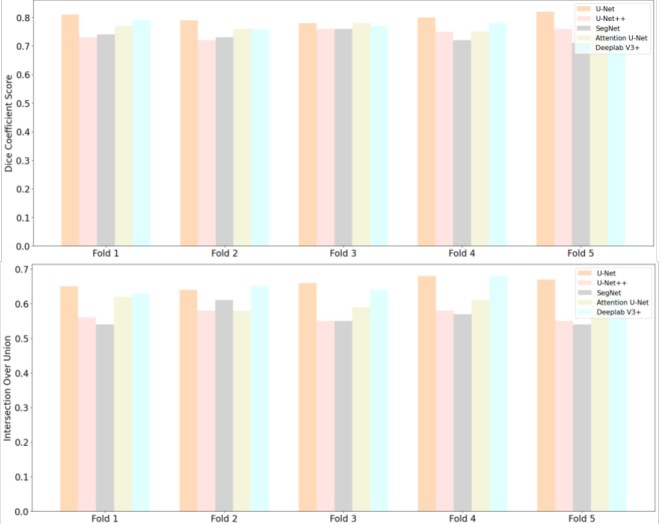

Fig. 3: Performance of Segmentation models across 5-fold cross-validation.

### C. Evaluation Metrics

The implementation of performance metrics is essential for determining a model's efficacy and constraints. Dice coefficient score (DCS), intersession over union (IoU) for segmentation and accuracy, F1 Score, precision, recall, and mean average precision (mAP) for object detection and penetration-aspiration detection are among the metrics used in the studies to assess the models. By calculating the degree of alignment between the model's output and the actual output at the pixel level, DCS evaluates a model's performance. Conversely, the IoU metric establishes how much the segmented and ground truth masks overlap [11], [12].

True positive (TP): The model accurately predicts the positive class. True negative (TN): The model accurately predicts the negative class. False positive (FP): The model incorrectly

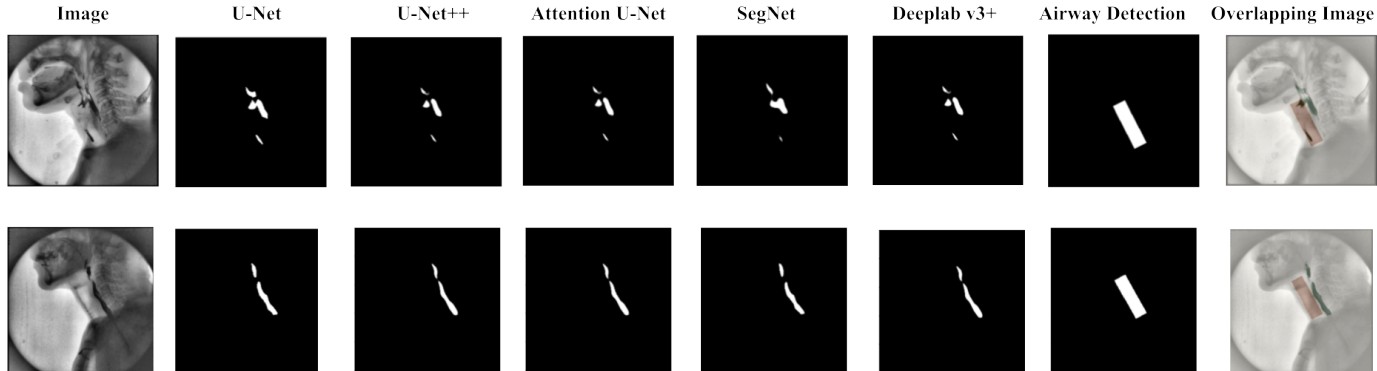

| Image | U-Net | U-Net++ | Attention U-Net | SegNet | Deeplab v3+ | Airway Detection | Overlapping Image |

Fig. 4: Bolus segmentation comparison: Abnormal patient results (top row) vs. normal patient results (bottom row).

forecasts the positive class. False negative (FN): The model incorrectly forecasts the negative class Precision quantifies the proportion of positive class predictions that are part of the positive class. Recall (also known as the true positive rate or sensitivity) measures how many positive class predictions were made out of all the positive examples in the dataset. The harmonic mean of recall and precision is known as the F1-score. The Mean Average Precision (mAP) is computed by determining the Average Precision (AP) for each class and subsequently averaging these values across all classes [16], [17].

$$DCS = \frac{2 * \text{Area of Overlap}}{\text{Total Area}} \quad (1)$$

$$IoU = \frac{\text{Area of Intersection}}{\text{Area of Union}} \quad (2)$$

$$Precision = \frac{\text{TP}}{\text{TP+FP}} \quad (3)$$

$$Recall = \frac{\text{TP}}{\text{TP+FN}} \quad (4)$$

$$F1 - Score = 2 * \frac{\text{Precision*Recall}}{\text{Precision+Recall}} \quad (5)$$

*D. Result Analysis*

We utilized a total of 2356 annotated image frames for training and validation of our model. To prevent patient frame repetition in the train and test sets, we split the images in an 80:20 ratio based on patient IDs. Both the segmentation models and YOLO-V8 underwent 5-fold cross-validation. Figure 2 shows the convergence of Yolo-V8 in airway detection. In Table II, we present a performance comparison of the segmentation models. The original U-Net model excelled in bolus segmentation, achieving a mean DCS of 0.80 and an

TABLE II: Bolus Segmentation performance comparison

| Model | DCS* | IoU* |
|---|---|---|
| **U-Net** | **0.80±0.01** | **0.66±0.01** |
| U-Net++ | 0.74±0.02 | 0.56±0.02 |
| Attention U-Net | 0.76±0.01 | 0.60±0.02 |
| SegNet | 0.73±0.01 | 0.56±0.02 |
| Deeplab V3+ | 0.77±0.01 | 0.64±0.02 |

* The results were presented in Mean±SD

TABLE III: Performance of Yolo-V8 in airway detection for 5 fold cross validation

| Fold | Precision | Recall | mAP50 | mAP50-95 |
|---|---|---|---|---|
| Fold 1 | 0.92 | 0.93 | 0.94 | 0.47 |
| Fold 2 | 0.93 | 0.92 | 0.95 | 0.57 |
| Fold 3 | 0.94 | 0.92 | 0.92 | 0.54 |
| Fold 4 | 0.91 | 0.85 | 0.92 | 0.54 |
| Fold 5 | 0.94 | 0.89 | 0.93 | 0.61 |
| **Mean±SD** | **0.92±0.01** | **0.90±0.03** | **0.93±0.01** | **0.54±0.05** |

IoU of 0.66. Table III showcases the 5-fold validation results for the YOLO-V8 model in airway detection, with average precision, recall, and mAP 50 scores of 0.92, 0.90, and 0.93, respectively. Figure 3 shows the performance of segmentation models across 5-fold cross-validation. Figure 4 illustrates the segmentation, and airway results from various segmentation models and YoloV8. For penetration-aspiration detection, we combined the outputs from different segmentation models with the YOLO-V8 model. Notably, the U-Net model, which performed best in bolus segmentation, also yielded the top result in penetration-aspiration detection. The macro average precision, recall, and F1 score were 0.90, 0.89, and 0.89, respectively. Figure 5 illustrates the pipeline's efficiency in penetration-aspiration detection.

TABLE IV: Penetration-Aspiration Detection

| AI Pipeline | Precision | Recall | F1 Score | Accuracy |
|---|---|---|---|---|
| **U-Net + Yolo-V8** | **0.90** | **0.89** | **0.89** | **0.89** |
| U-Net++ + Yolo-V8 | 0.86 | 0.85 | 0.85 | 0.86 |
| Attention U-Net + Yolo-V8 | 0.90 | 0.88 | 0.88 | 0.88 |
| SegNet + Yolo-V8 | 0.85 | 0.84 | 0.84 | 0.84 |
| Deeplab V3+ + Yolo-V8 | 0.87 | 0.86 | 0.86 | 0.86 |

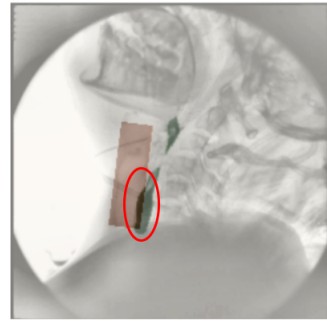

Fig. 6: Pipeline misclassified the normal case as a penetration-aspiration case.

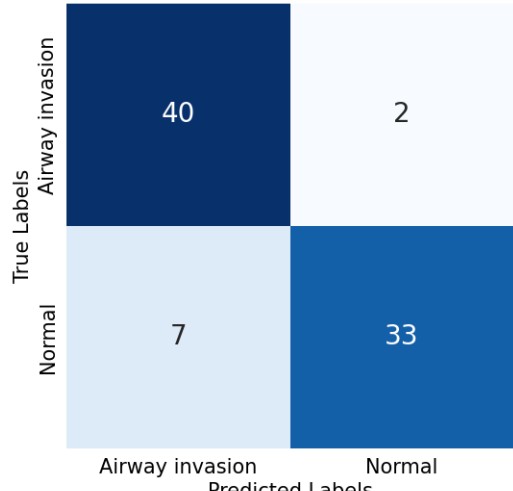

Fig. 5: Confusion matrix of prediction using AI pipeline consists of U-Net and Yolo-V8 model.

## V. DISCUSSION

In this article, we propose an AI pipeline for penetration-aspiration detection, encompassing bolus segmentation and airway detection. Both the bolus and airway play critical roles in the swallowing process, and analyzing these components holds significance for penetration-aspiration detection. To enhance airway invasion detection, we incorporated bolus and airway tracking modules into our pipeline. For bolus segmentation, we employed various state-of-the-art models, including U-Net++, U-Net, SegNet, Attention U-Net, and Deeplab V3+. Among these, the U-Net model exhibited the best performance. In constructing the training dataset, we observed that one-third of the annotation frames are sufficient for achieving results comparable to the state-of-the-art studies on bolus segmentation.

Previous research has explored bolus segmentation. For instance, Ariji et al. [11] achieved a DCS of 0.95 in bolus segmentation, albeit with a small number of patients. Other

studies by Shaheen et al. [12] and Li et al. [13] reported DCS scores of 0.67 and 0.81, respectively. Our U-Net model also achieved a DCS score of 0.80, aligning with existing research. Notably, our study is the first to detect airways in VFSS images for penetration-aspiration detection. Leveraging the YOLO-V8 model, we fine-tuned it to achieve an impressive 0.93 mAP in airway detection.

The bolus and airway tracking within VFSS clips hold promise for penetration-aspiration detection. Our pipeline capitalizes on bolus segmentation and airway detection, a novel approach to decision-making. With a macro average F1 score of 0.89, our AI pipeline performs comparably to existing studies. Lee et al. [16], Kim et al. [17], and Reddy et al. [18] have explored DL for penetration-aspiration detection, achieving accuracies of 93.2%, 94.7%, and 90%, respectively. Our pipeline also achieves comparable results, with an 89% accuracy.

However, our AI pipeline has limitations. It relies on the performance of the bolus segmentation model and airway detection model. In some cases, when creating rectangle bounding box masks for airways, portions of the food path may be covered, leading to the misclassification of normal patients as at-risk (Figure 6). Additionally, in abnormal patient swallowing videos, a small number of overlapping frames can lead to misclassification of penetration-aspiration patients as normal. These overlapping frames predominantly occur during the pharyngeal phase of the VFSS clip. To enhance performance, integration of the pharyngeal frame identification module into our pipeline is needed. Additionally, it's important to note that our pipeline has been trained and validated on our hospital's VFSS data. To improve its generalizability, we should consider training and validating it using multicenter data.

## VI. CONCLUSION

This article presents the first AI pipeline incorporating bolus segmentation and airway detection to detect penetration-aspiration. We curated the VFSS dataset to facilitate bolus segmentation, airway detection, and penetration detection. Remarkably, our experiments demonstrate that using only one-third of the annotated frames from VFSS clips, we achieve comparable results to state-of-the-art bolus segmentation and

penetration-aspiration detection methods. Notably, our study pioneers airway detection in VFSS images. We fine-tuned YOLO-V8 for airway detection, achieving a state-of-the-art mAP50 of 0.93. This implies that our pipeline leverages the features of both the bolus and the airway, which are two critical components involved in penetration-aspiration. These promising results have significant implications for supporting clinical decision-making. Moving forward, our research aims to further refine the AI pipeline and deploy clinical translational tools for assessing penetration-aspiration risk in the penetration aspiration scale.

## ACKNOWLEDGMENT

This study was supported through the AMRITA Seed Grant (Proposal ID: ASG2022-008)

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
