# OpenReview forum: "Artificial Intelligence-Driven Penetration-Aspiration Detection in Dysphagia Patients Using Fluoroscopic Videos"
_IEEE.org/EMBS/BHI/2024/Conference — IEEE BHI'24_

### Official Review · Reviewer_sQV7 · 2024-08-02
**Well-written paper**

**Overall Rating:** 7
**Confidence:** 4

**Other Quality Metrics:**

(a) Clarity of writing: Excellent;
(b) Clinical Significance: Great;
(c) Methodological Novelty: Good;
(d) Experiments and Results: Great

**Questions For The Authors:**

- The purpose of the state-of-the-art papers that only segmented the bolus is not clear. Could this be used also for airway invasion detection?

**Strengths:**

The paper is well-written and all explanations are clear. The results are promising, despite the limitations presented.

**Summary Of The Paper:**

The paper presents an AI pipeline for airway invasion detection including bolus segmentation, airway detection and classification.

**Weaknesses:**

When comparing the results with state-of-the-art papers, the novelty of the current paper should be highlighted as it is in section II.

---

### Official Review · Reviewer_NAjQ · 2024-08-06
**AI based penetration-aspiration detection, with bolus segmentation and airway detection**

**Overall Rating:** 7
**Confidence:** 4

**Other Quality Metrics:**

(a) Clarity of writing; Great
(b) Clinical Significance; Great
(c) Methodological Novelty; Good
(d) Experiments and Results: Good

**Questions For The Authors:**

Nice work on addressing a very specific and important problem area. I have a few questions to better understand your study and its future direction:

1. Data Source:
Why did you choose to use data from only one hospital for your study?
Are there specific reasons or constraints that limited your ability to include data from multiple or more diverse sources?
How do you think using a single hospital dataset might have affected your results, and what steps are you considering to mitigate any potential biases?
2. Next Steps:
What are the next steps for your study?
Do you have plans to expand the dataset or validate your AI pipeline with data from other hospitals or regions?
Are there any ongoing or planned efforts to address the limitations identified in your current study?
3. Misclassification Issues:
How are you addressing the misclassification of normal cases as penetration-aspiration cases?
Are there specific strategies or model adjustments you are implementing to improve the accuracy and reduce false positives?
Can you provide more details on the measures you are taking to enhance the specificity of your AI pipeline?

**Strengths:**

1. Focus on a Specific Problem:
The study addresses a highly specific and clinically significant problem: the assessment of penetration-aspiration risk in patients with dysphagia using VFSS. This focus allows for a detailed and targeted approach to improving diagnostic accuracy and consistency.
2. Novel AI Pipeline:
The introduction of a novel AI pipeline that combines bolus segmentation and airway detection is a major strength. This combinational assessment provides a more comprehensive and precise analysis of VFSS videos compared to existing methods.
3. Comparison with Existing Solutions:
The study includes a robust comparison with existing AI approaches, highlighting the improvements and innovations introduced by the proposed pipeline. This comparative analysis underscores the advancements made in addressing the limitations of previous methods.
4. Thorough Methodological Analysis:
The paper presents a detailed and systematic analysis of the entire process, from data curation and annotation to model development and evaluation. This thoroughness enhances the credibility and reproducibility of the findings.
5. Clinical Relevance and Efficiency:
The proposed pipeline not only improves diagnostic accuracy but also reduces the manual annotation burden on clinicians, saving time and resources. This efficiency is a significant advantage in clinical practice.
6. Acknowledgement of Limitations and Future Work:
The authors openly acknowledge the limitations of their approach and suggest next steps for future research. This transparency is commendable and provides a clear direction for further improvements and validation of the pipeline.

**Summary Of The Paper:**

Objective: This article introduces a novel AI pipeline aimed at improving the accuracy and consistency of penetration-aspiration risk assessment in VFSS. The proposed approach leverages bolus segmentation and airway detection to provide a comprehensive frame-wise interpretation of penetration-aspiration risk.

Methodology:

Current Limitations: Existing AI methods for VFSS analysis typically depend on manual frame selection and fail to adequately consider the clinical significance of both the bolus and the airway.
AI Pipeline Development: To address these limitations and the challenges posed by varying airway orientations, an automated AI pipeline was developed. This pipeline tracks both the bolus and the airway throughout VFSS videos.
Data Curation and Annotation: Due to the absence of established benchmarks, a VFSS dataset was curated, and one-third of the frames from 82 VFSS clips obtained from 40 patients were annotated.
Model Comparison and Fine-Tuning: Various segmentation models were compared for bolus segmentation, and an object detection model was fine-tuned for airway detection.
Results:

Performance Metrics: The AI pipeline achieved a dice score of 0.80, a mean average precision of 0.93, and an accuracy of 89% in bolus segmentation, airway detection, and penetration-aspiration detection.
Efficiency: The pipeline demonstrated effective training capabilities even with a limited number of annotated frames, significantly reducing the burden of manual annotation for clinicians.
Conclusion: The proposed AI pipeline shows promising results in assisting clinicians with the assessment of penetration-aspiration risk in dysphagia patients, potentially improving the accuracy and consistency of VFSS interpretations.

**Weaknesses:**

1. Limited Dataset:
One significant weakness of the study is the limited dataset used. The authors utilized a total of 2,356 annotated image frames from 82 VFSS clips obtained from 40 patients. While the results are promising, the small dataset may not be fully representative of the broader patient population, potentially affecting the generalizability of the findings.
2. Misclassification Issues:
The pipeline demonstrated instances of misclassifying normal cases as penetration-aspiration cases. This type of false positive can lead to unnecessary concern and further diagnostic procedures for patients, highlighting a critical area for improvement in the model's accuracy and specificity.
3. Lack of General Dataset:
The study relies on VFSS data from a specific hospital, which may introduce bias and limit the applicability of the results to other clinical settings. The absence of a more diverse and general dataset means that the AI pipeline might not perform as well in different hospitals or with different patient demographics.
4. Potential Overfitting:
Given the limited size and specific nature of the dataset, there is a risk of overfitting, where the model performs well on the training data but fails to generalize to new, unseen data. This issue is particularly pertinent in medical AI applications where robustness and generalizability are crucial.
5. Lack of External Validation:
The study does not include external validation with independent datasets from other institutions. External validation is essential to demonstrate the pipeline's effectiveness and reliability across different clinical environments and patient populations.

---

### Decision · Program_Chairs · 2024-09-23

Accept